# Aripiprazole reduces liver cell division

**Tinkara Pirc Marolt[1], Barbara Kramar[1], Klara Bulc Rozman[1], Dušan Šuput[1], Irina Milisav** [1,2]*

**1** Institute of Pathophysiology, Faculty of Medicine, University of Ljubljana, Ljubljana, Slovenia, **2** Laboratory of Oxidative Stress Research, Faculty of Health Sciences, University of Ljubljana, Ljubljana, Slovenia

* irina.milisav@mf.uni-lj.si

**Data Availability Statement:** All relevant data are available from the University of Ljubljana database via the following URL: https://repozitorij.uni-lj.si/IzpisGradiva.php?id=121367.

**Funding:** This work was supported by the Slovenian Research Agency in the form of funding

## Abstract

Effects of aripiprazole on dopamine regulation are being tested as a treatment for patients with a dual diagnosis of schizophrenia and addictions, often cocaine dependence. Aripiprazole has one of the fewest side-effects among the second-generation antipsychotics. Nevertheless, severe aripiprazole hepatotoxicity was reported in persons with a history of cocaine and alcohol abuse. Here we report that therapeutically relevant aripiprazole concentrations, equal to laboratory alert levels in patients' serum, reduce the rate of hepatocytes' division. This could be an underlying mechanism of severe liver injury development in the patients with a history of alcohol and cocaine abuse, the two hepatotoxic agents that require increased ability of liver self-regeneration. Monitoring liver functions is, therefore, important in the cases when aripiprazole is co-prescribed or used with drugs with potential hepatotoxic effects.

## Introduction

Aripiprazole (ARI) has fewer reported metabolic side-effects compared to some other frequently used second-generation antipsychotics. Nevertheless, there are reports of individuals that developed side-effects, like hypertension [1], diabetes mellitus [2], rhabdomyolysis and elevated liver enzymes [3]. Liver complications are generally rare, although they can include even severe complications, like drug-induced liver injury (DILI). For example, hepatitis after six weeks of ARI monotherapy was reported in a 52-year old patient with a 20-year history of increased alcohol consumption [4]. A severe aripiprazole-caused liver injury occurred in a 28-year female patient, only 21 days after she started taking 10 mg ARI and three times 2.5 mg lorazepam per day for 12 days. The ARI dose was after that increased to 20 mg per day up to the onset of the liver adverse effects [5]. Paliperidone was then prescribed instead of the ARI, while lorazepam treatment remained unchanged. The patient had previously no known medical conditions, even no psychiatric disease, as she was only first diagnosed 21 days before the DILI development. She reported inhaling cocaine once per month.

Alcohol or illicit drug dependence is estimated in up to 50% of patients with schizophrenia [6, 7], and is 4.6-times higher than in the population without this disease [8]. As comorbidities like cocaine and alcohol abuse exacerbate schizophrenia symptoms, and ARI has dopamine-modulating properties, several clinical trials were initiated to test whether ARI could reduce the addiction. After an encouraging pilot study, in which reduced cocaine amounts were

awarded to DS, IM, adn KBR (research core
funding no. P3-0019) and TPM. This work was
also partially supported by H2020 Excellent
Science in the form of project funding awarded to
BK and IM (H2020-MSCA-ITN:721236
TREATMENT). The funders had no role in study
design, data collection and analysis, decision to
publish, or preparation of the manuscript.

**Competing interests:** The authors have declared
that no competing interests exist.

**Abbreviations:** ARI, aripiprazole; DILI, drug-
induced liver injury; OLA, olanzapine.

found in the urine of 6 out of 10 enrolled male participants from the third week of ARI treat-
ment [9], a larger trial on 44 persons (22 per ARI group) detected no reduction in the urine
cocaine levels, only decreased cocaine craving from the week 6–8 of the trial [10]. There are
conflicting reports that ARI was not useful for treating the cocaine dependence [11, 12],
reports with inconclusive results [13] and those describing a slight increase in cravings in for-
mer cocaine users who have achieved abstinence [14]. Likewise, there is no clear conclusion
on the ARI prevention of alcohol dependence [15–17], while ARI was not efficacious for the
treatment of methamphetamine addiction and two trials were terminated because of adverse
reactions [18, 19].

DILI is a potentially life-threatening adverse response to any drug [20]. It accounts for
more than 50% of cases of liver failure [21] and it is the most frequent reason for drug with-
drawal from the market [22]. Idiosyncratic DILI is one of the most challenging liver disorders
faced by hepatologists, despite its occurrence in only a small proportion of exposed individuals
[23]. Its onset is unpredictable, usually not dose-related above a drug threshold and with vari-
able latency from days to weeks. Currently there is no simple way for adequate prediction of
DILI in laboratory models used for preclinical drug testing. The results of a retrospective anal-
ysis of marketed pharmaceuticals implied that cytotoxicity assays in human hepatocellular car-
cinoma HepG2 cells were not sensitive enough to detect human hepatotoxic drugs [24].
Nevertheless, there is a single report that oral treatment of rats with therapeutic and maximal
therapeutic doses of ARI and carbamazepine or ARI and fluvoxamine for eight weeks resulted
in severe liver injury [25].

Here we report a significant growth decrease of hepatocyte model cells Fao and of immor-
talized hepatocytes that were continuously exposed to ARI for about six and two weeks, respec-
tively. Their growth rate was compared to either untreated controls or cells treated with
another second-generation antipsychotic, olanzapine (OLA). Lower cell numbers that were
regularly detected at longer ARI treatments resulted from decreased cell division, the conse-
quence of which may result in reduced liver regeneration. Therefore, hepatic damage should
be monitored when ARI is prescribed to patients with a history of ingesting hepatotoxic sub-
stances, like alcohol and illegal drugs.

## Materials and methods

All reagents were purchased from Sigma-Aldrich (Merck), unless otherwise stated.

### Cell cultures

The rat hepatoma cell line Fao (ECACC, 89042701) was grown at 37˚C in a humidified atmo-
sphere with 5% $CO_2$, in Coon's F-12 Modified Liquid Medium (MBC-F0855) supplemented
with 10% fetal bovine serum (Gibco, 10270–106), 1% penicillin/streptomycin (Gibco, 15140–
122) and 1% GlutaMAX supplement (Gibco, 35050038). The seeded cells were rested for 24
hours before a single treatment for 24 hours with ARI (PHR1784, CAS Number: 129722-12-9)
or OLA (PHR1825, CAS Number: 132539-06-1), both dissolved in 0.12% DMSO (Acros
Organics, 67-68-5) final concentration. For long-term treatment, several parallels from differ-
ent batches of cells were continuously propagated in media with selected concentrations of
antipsychotics or vehicle control (untreated control) for four weeks before they were used in
experiments. No experimental difference was observed between four- and eight- weeks treated
cells. All experiments were completed after eight weeks of ARI/OLA treatment.

For cytotoxicity and caspase activity assessments, 40.000 cells/well were seeded in 96-well
flat-bottom cell culture plates. 12-well cell culture plates (400.000 cells/well) and T25 flask (2.4
million cells/flask) were used for caspase activity and cytometric analysis and 15 mm coverslips

(100.000 cells/coverslip) for immunocytochemistry. Cells were seeded in 48-well cell culture plates (22.000 cells/well) for senescence-associated β-galactosidase (SA-β-gal) activity. All assays were started 24 hours after seeding, except for the SA-β-gal assay, which was performed 48 hours after seeding.

Immortalized mouse neonatal hepatocytes were kindly provided by Dr Ángela M. Valverde (Instituto de Investigaciones Biomedicas 'Alberto Sols', CSIC, Madrid, Spain) [26, 27]. The cell line was grown at 37˚C in a humidified atmosphere with 5% $CO_2$, in a high glucose DMEM (4.5 g/L glucose + L-glutamine, 11965–092) supplemented with 10% fetal bovine serum (Gibco, 10270–106) and 1% penicillin/streptomycin (Gibco, 15140–122). Cells were seeded in T25 flasks (100.000 cells/flask) and treated with ARI and OLA for two weeks.

## Cell viability assays

Cytotoxicity was evaluated with 0.04 mg/mL Neutral Red dye (3-Amino-7-dimethylamino-2-methylphenazine hydrochloride, N4638) dissolved in the growth medium for two hours and washed with PBS. A reaction mix, composed of 50% ethanol and 1% glacial acetic acid in water, was added to cells before the absorbance measurement at 550 nm (PerkinElmer, Victor 1420–050 spectrophotometer).

Dehydrogenase activity was measured by the 2-hour accumulation of 0.5 mg/mL Thiazolyl Blue Tetrazolium Bromide (MTT, M5655) dissolved in the growth medium; the accumulated crystals were released by DMSO and absorbance measured at 550 nm (PerkinElmer, Victor spectrophotometer 1420–050).

Released lactate dehydrogenase (LDH) was measured from the culture supernatants with LDH Cytotoxicity Detection Kit (Takara, MK401) according to the manufacturer's instructions.

## Caspase activity

Activities of caspase-3/7 and -9 were measured in CCLR buffer with Caspase-Glo® kit (Promega, Madison, WI, USA; Caspase-3/7 –G811A, Caspase-9 –G8211) as described before [28]. Protein concentrations were measured with Pierce 660 assay (Thermo Fisher Scientific, 22660). Samples were introduced in a caspase-activity reagent mix to measure luminescence by GloMax® 20/20 Luminometer (Promega, E5311).

## Flow cytometry

Cell concentration, viability, cell proliferation and cell cycle were assessed with Muse™ Cell Analyser (Millipore Corporation, now Guava Muse Cell Analyser, Luminex) using Count & Viability (MCH100102), Ki67 Proliferation (MCH100114) and Cell Cycle (MCH100106) kits for fluorescence detection by microcapillary flow cytometry.

## RNA isolation and reverse-transcription quantitative polymerase chain reaction analysis (RT-qPCR)

Total RNA was isolated with TRI reagent (T9424) and reverse transcribed using the High capacity cDNA reverse transcription kit (Applied Biosystems, 4368814) with added RNase inhibitor (Applied Biosystems, N8080119). PCR reactions ($\leq$ 100 ng cDNA/reaction) were run in duplicates using TaqMan Universal Master Mix II, with uracil-N-glycosylase (Thermo Fisher Scientific, 4440038) and quantitated using the 7500 Real-Time PCR System with SDS software (v1.3.1, Applied Biosystems). SDS software was used to set baselines and to determine threshold cycles (CT). PCR efficiency (E) of each assay was determined using LinRegPCR software [29,

30]. Expression of target genes was calculated relative to expression of a reference gene for 18S rRNA (*Rn18s*) using the following equation: target/reference = (E(reference)$^{Ct(reference)}$)/(E(target)$^{Ct(target)}$). Results were then expressed as fold change compared to the untreated sample. The following TaqMan probes labelled with the FAM dye (Thermo Fisher Scientific) were used: the reference gene *Rn18s* (Rn03928990_g1), *p53* (Rn00755717_m1), *Gadd45α* (Rn00577049_m1), *p27* (Rn00582195_m1), *p21* (Rn00589996_m1) and *p16* (Rn00580664_m1).

### Immunocytochemistry and senescence

Cells grown on coverslips for 72 h, were fixed for 10 min in ice-cold 4% paraformaldehyde in PBS and permeabilized with 0.1% Triton X-100 in 4% paraformaldehyde/PBS for 10 minutes at room temperature. Coverslips were incubated with 10% goat serum in PBS (Gibco 16210064) for one hour at 37˚C followed by 1-hour incubation with the primary antibody (1:300, Ki67 Monoclonal Antibody (SolA15), FITC, eBioscience™, Invitrogen, 11-5698-82). Cell nuclei were stained with Hoechst 33342 (Invitrogen, H1399) diluted 1:5000, washed with PBS and mounted with ProLong™ Diamond Antifade Mountant (Thermo Fisher Scientific, P3695).

The senescence-associated β-galactosidase activity was investigated on the 5$^{th}$ day after cell seeding using SA-β-gal staining kit (Cell Signaling Technology, 9860), according to the manufacturer's instructions. Images of cells in PBS were acquired at room temperature (Olympus IX81F microscope with Olympus DP71 camera). Six vision fields for each treatment in three biological replicates were analyzed.

### Statistical analysis

Statistical analysis was performed with GraphPad Prism 8.4.3, which has an inbuilt algorithm to test the equality of variances from medians, the Brown-Forsythe test. In the case of equal variances, one-way ANOVA was used for analyses of data with more than 2 groups. Then, treated samples were compared to an untreated control with Dunnett's test. Statistical parameters, including the number of biological replicates (*n*), are noted in figure legends and S5 Fig. All data on graphs are presented as means ± SD.

## Results

Primary hepatocytes survive in cell cultures for only about a week and although there is a transition from the G0 into the G1 phase induced at isolation, they do not divide in cell culture [28, 31, 32]. Because of reports that human hepatocyte carcinoma, HepG2 cells, were not sensitive enough to detect hepatotoxic effects of drugs in preclinical studies [24], we used rat hepatoma cells Fao, which divide in cell cultures and are suitable for metabolic studies [33–35], and confirmed the crucial results on immortalized mouse hepatocytes [26, 27].

### Long-term ARI treatment reduces the number of cells

Reduced cell number was first observed after two weeks when the Fao cells were continuously treated with 6 μM ARI. This reduction was proportional to the concentration of ARI. It was also detected from three weeks at 2.23 μM ARI, the drug concentration equivalent to the laboratory alert level, i.e. the ARI concentration in the patient's serum with reported side-effects [36] (Fig 1A and 1B). The same concentrations of ARI also reduced growth of immortalized hepatocytes from neonatal mice within 1–2 weeks of treatment. Dehydrogenase activity, which is often used as a cell viability test (MTT), was significantly reduced in the ARI treated cells. However, dehydrogenase activity decreased even after a single ARI treatment (Fig 1C) and

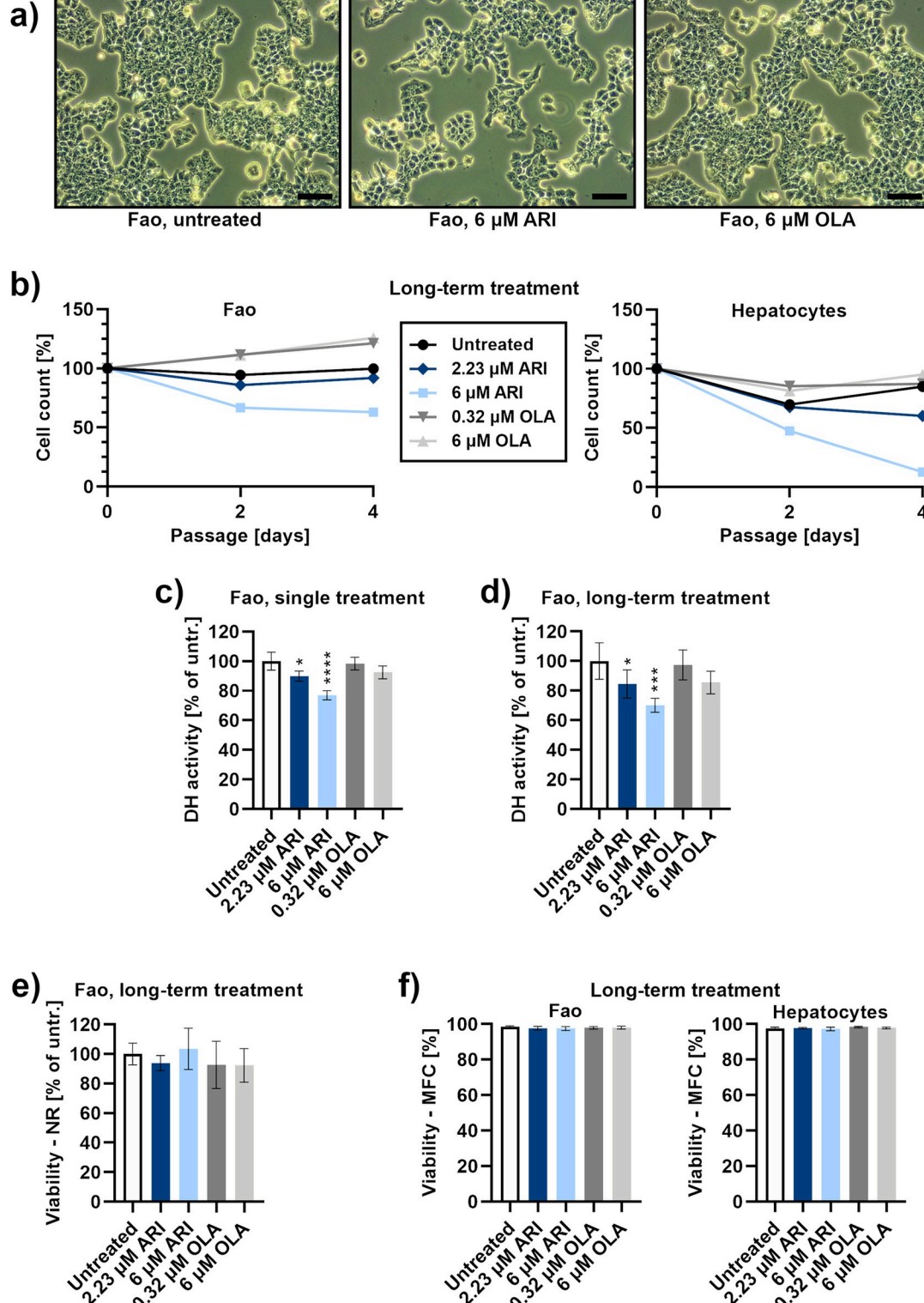

**Fig 1. Liver cell survival. a)** Phase-contrast micrographs of Fao cells after long-term treatment: untreated control (left), 6 μM ARI (middle) and 6 μM OLA (right). Micrographs were taken at day 4 of the cell count time-course in Fig 1B. Scale bars: 50 μm. **b)** Cell count time-course, using microcapillary flow cytometry. Left side: Fao cell concentrations at the three time points of cell passaging (every second day) are shown as the percentage of cell concentration of the first passage (day 0); *n* = 5 (biological replicates). Right side: the concentrations of immortalized hepatocytes; *n* = 2. Representative flow cytometric dot plots are in S1

and S2 Figs. Dehydrogenase (DH) activity, evaluated with MTT assay in **c)** single treated (*n* = 4) and **d)** long-term treated cells (*n* = 5). **e)** Cytotoxic effect, evaluated with Neutral red assay, in long-term treated cells; *n* = 4. **f)** Viability of long-term treated cells during passaging, using microcapillary flow cytometry (MFC); left side: Fao cells, *n* = 5. Right side: immortalized hepatocytes, *n* = 6. Flow cytometric dot plots are in S1 and S2 Figs. Data are presented as the mean ± SD and analysed with one-way ANOVA followed by Dunnett's test. *P ≤ 0.05, ***P ≤ 0.001, ****P ≤ 0.0001.

remained lower during long-term ARI treatments (Fig 1D). All cells equally retained the ability of live cells to accumulate a neutral red (NR) dye in lysosomes (Fig 1E). Equal viability among all long-term treated cells was also confirmed by microcapillary flow cytometry, in both, Fao cells and immortalized hepatocytes (Fig 1F). This implies that cell survival is unchanged in all treated cells during the long-term treatment.

## ARI does not induce hepatotoxicity

Extracellular concentrations of lactate dehydrogenase (LDH), a cytoplasmic enzyme that is released upon the plasma membrane damage, did not significantly differ among each other or to the amount released from the untreated control after a single and long-term ARI or OLA treatments (Fig 2A and 2B). Also, no extensive apoptosis was initiated, as no caspase activity was markedly induced at single or long-term ARI treatments (Fig 2C–2F). Cell damage and caspase activation cannot account for the lower number of cells in the ARI treated samples.

## Reduced proliferation of ARI treated cells

Except for the samples used to measure decreasing amounts of cells in Fig 1B, the number of cells was kept equal in all groups to avoid the differences that could have occurred due to cell density variability (Fig 3A). The expression of nuclear Ki67 protein, an established marker of cell proliferation [37], was specifically lower upon the 6 μM ARI long-term treatment. No differences were observed with a single treatment. Expression of Ki67 was investigated by microcapillary flow cytometry (Fig 3B and 3C) and immunocytochemistry (Fig 3D), with latter detecting the smallest amount of green signal in 6 μM ARI long-term treated cells which accounted for proliferative cells.

Reduced division of ARI treated cells was confirmed by cell cycle analysis, as significantly more long-term ARI treated cells remained in the interphase or in G1 phase (G0/G1 phase), compared to the OLA or untreated cells and single treated cells (Fig 4). Gene expression of tumour protein 53 (*p53*) and growth arrest and DNA damage-inducible protein (*Gadd45α*), which can be induced by p53, was determined (Fig 4E and 4F). An overall statistically significant difference was calculated between the *Gadd45α* expression levels of all samples (P = 0.0382, one-way ANOVA). As the differences between the samples were small, no significant difference in gene expression was observed between any treated sample and the untreated control (Dunnett's test, S5 Fig). We then examined the levels of cell cycle inhibitors of the p16/cyclin-dependent kinase (CDK) 4: cyclin D/retinoblastoma pathway. Expression of cell cycle inhibitors *p21* and *p27* did not significantly differ between the ARI treatments and untreated cells (Fig 4G and 4H). The small difference in expression of *p27* between all treatments resulted in an overall statistically significant difference (P = 0.0289, one-way ANOVA), but there was no significant difference between any treated sample and the untreated control (Dunnett's test, S5 Fig).

The possibility of the cell cycle arrest due to senescence was then investigated, as the cell cycle arrest in G1/S transition is a typical feature of senescent cells [38]. No intense β-galactosidase labelling was observed in cells and there was a small number of labelled cells. The largest number of senescent cells were among the ARI treated cells and their percentage was

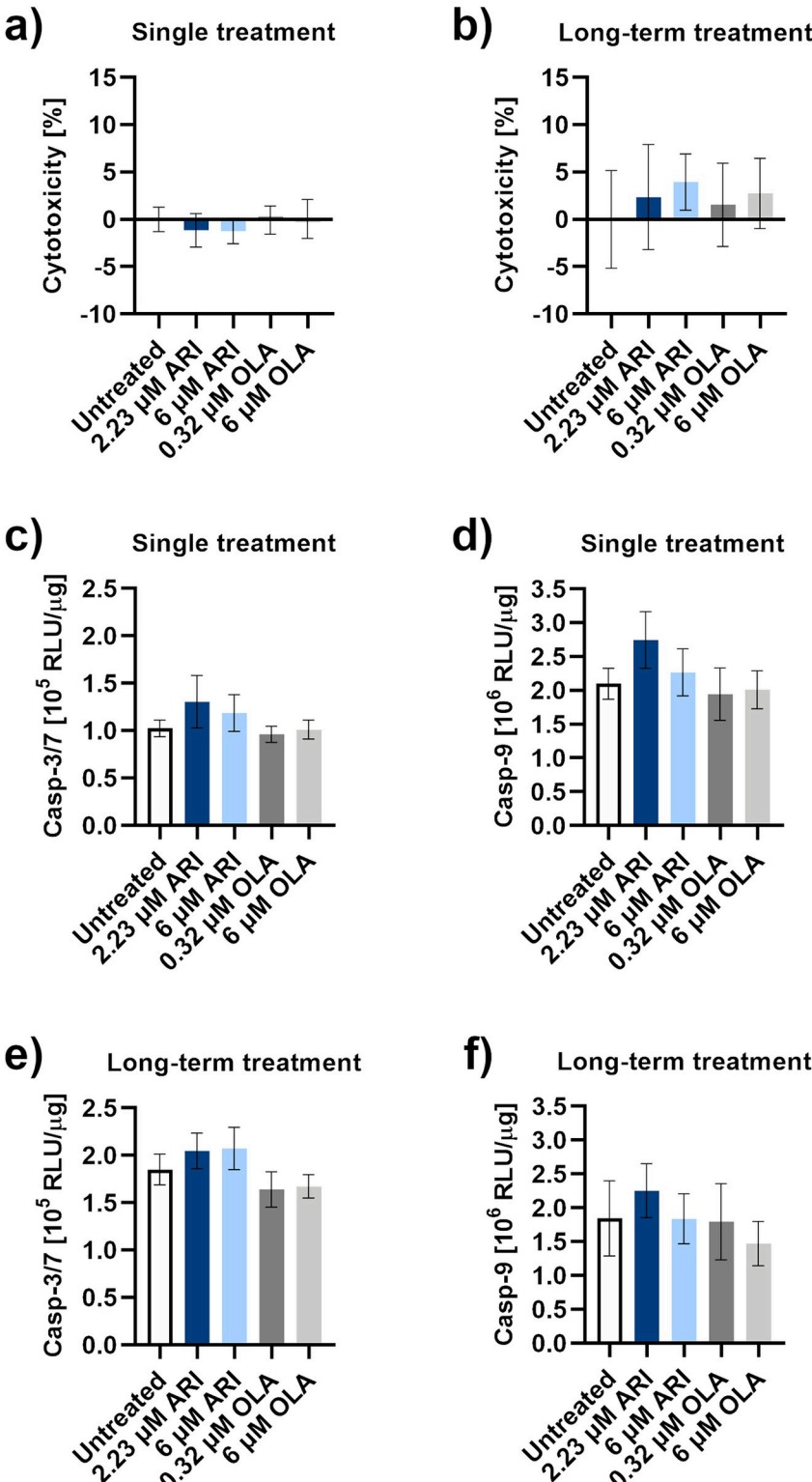

**Fig 2. Hepatotoxicity.** LDH release in **a)** a single treated (*n* = 4) and **b)** long-term treated Fao cells (*n* = 8). **c)** Caspase-3/7 and **d)** caspase-9 activities in cells upon a single treatment; *n* = 4. **e)** Caspase-3/7 and **f)** caspase-9 activities in Fao cells during long-term treatments; *n* = 4. Data are presented as the mean ± SD and analysed with one-way ANOVA. RLU: relative luminescence units.

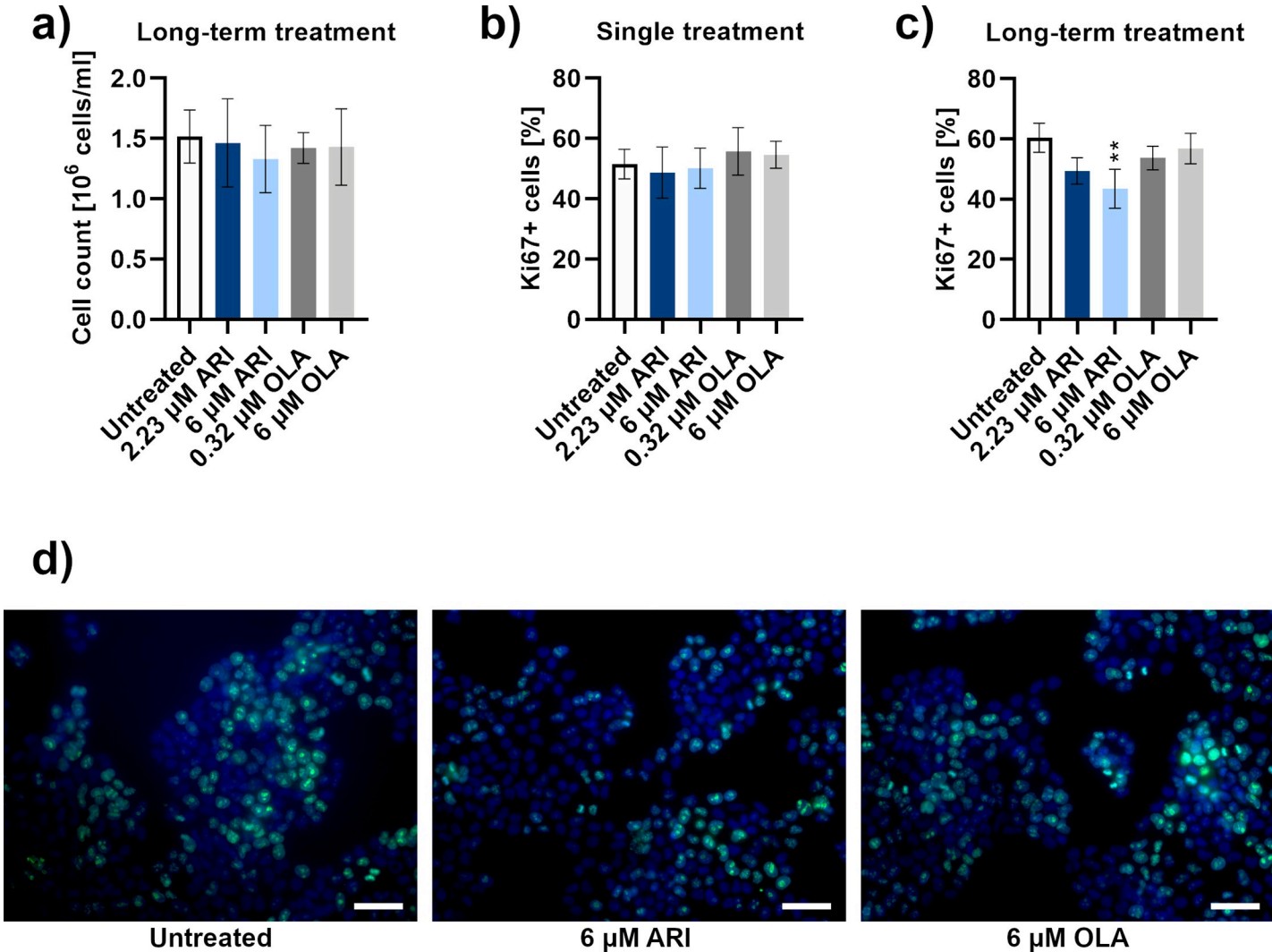

**Fig 3. Proliferation. a)** Cell count of long-term treated Fao cells by microcapillary flow cytometry with adjusted seeding to result in equal cell numbers ($n$ = 5). Proliferating Ki67+ cells (% of Ki67-positive cells of all cells in the sample) in **b)** single treated ($n$ = 6) and **c)** long-term treated ($n$ = 3) Fao cells, detected by microcapillary flow cytometry. Flow cytometric dot plots are in the S3 Fig. **d)** Fluorescent micrographs of Fao cells after long-term treatments were stained with Ki67 antibody (green signal) and Hoescht (blue signal): untreated control (left), 6 µM ARI (middle) and 6 µM OLA (right). Scale bars: 50 µm. Data are presented as mean ± SD and analysed with one-way ANOVA followed by Dunnett's test. **P $\leq$ 0.01.

concentration dependent. Statistical significance was reached at only 6 µM ARI at about 5% of senescent cells (Fig 5).

## Discussion

Assessment of survival, toxicity, proliferation, cell cycle and senescence in the ARI treated liver model reveals a decreased ability of hepatic cell regeneration (Table 1). The cells are as viable and non-apoptotic as in the untreated control or OLA treated samples, but they do not proliferate at the same rate. Senescence cannot account for reduced proliferation, as the senescent cell increase is minute and is smaller than the increase of non-replicating cells in both ARI samples. To check whether DNA damage could play a role in the slower growth of ARI treated samples, we determined the expression of *Gadd45α* gene, encoding the protein that induces

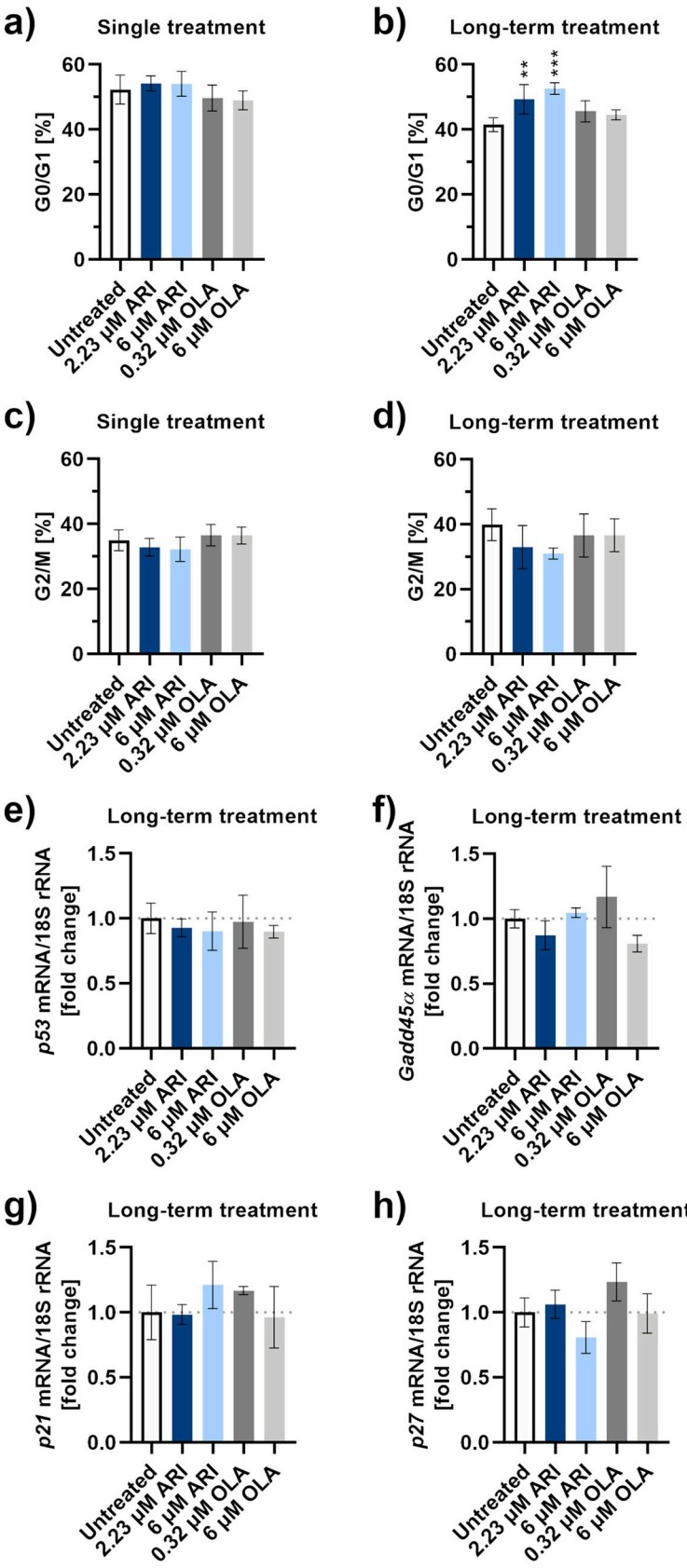

**Fig 4. Cell cycle progression and gene expression.** Percentage of Fao cells in the G0/G1 phase of the cell cycle compared to all cells in the sample during **a)** single-treatment and **b)** long-term treatment, detected by microcapillary flow cytometry; *n* = 4. Percentage of Fao cells in the G2/M phase of the cell cycle compared to all cells in the sample during **c)** single-treatment and **d)** long-term treatment, detected by microcapillary flow cytometry; *n* = 4. Flow cytometric dot plots are in the S4 Fig. Gene expression levels of cell cycle regulators, **e)** *p53*, **f)** *Gadd45α*, **g)** *p21* and **h)** *p27*, and the reference gene for 18S rRNA were monitored by RT-qPCR. Data are presented as the mean ± SD and analysed with one-way ANOVA followed by Dunnett's test. **P ≤ 0.01, ***P ≤ 0.001.

cell cycle arrest upon the DNA damage. The *Gadd45α* levels do not differ significantly among any combination of ARI- and OLA- treated and untreated cells. Gene expression levels of its upstream effector, the protein p53, are also similar among all groups of treated cells. In addition to mediating the induction of Gadd45α [39], p53 is involved in repression of cell cycle genes by transactivation of *p21* [40]. The expression of *p21* is also unchanged among the ARI and other treatments and so is the expression of *p27*; both proteins belong to the same group

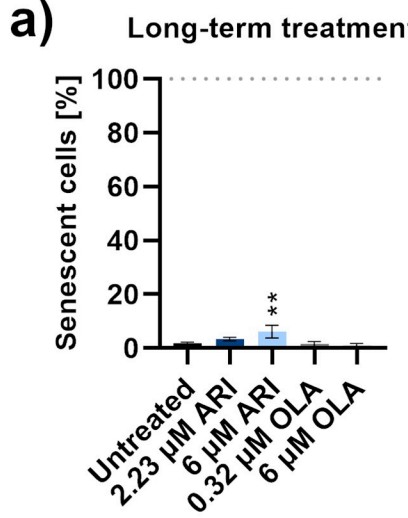

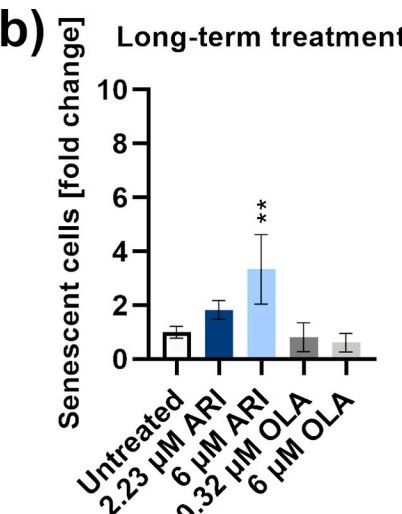

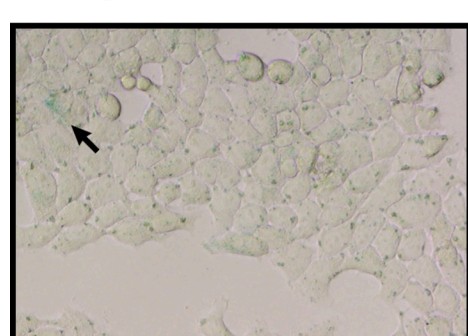

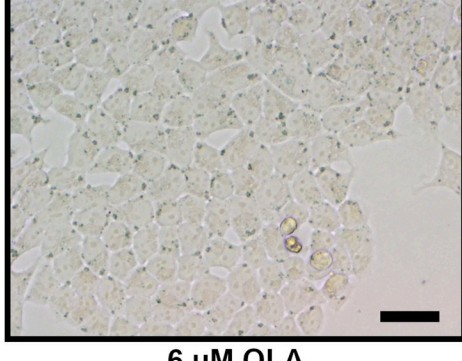

**Untreated**     **6 μM ARI**     **6 μM OLA**

**Fig 5. Senescence. a)** Percentage of senescent Fao cells according to β-galactosidase activity after long-term treatment; *n* = 3. **b)** Data from a) are shown as the percentage of senescent cells relative to the untreated control cells. **c)** Micrographs of Fao cells stained for β-galactosidase activity after long-term treatment: untreated control (left), 6 μM ARI (middle) and 6 μM OLA (right). Scale bars: 30 μm. Data are presented as mean ± SD and analysed with one-way ANOVA followed by Dunnett's test. **P ≤ 0.01.

**Table 1. Summary effects of the ARI long-term treatment of Fao cells.**

| | Viability | | | | |
|---|---|---|---|---|---|
| | Cell count time-course (MFC, Fig 1B) | Viability (NR, Fig 1E) | Viability (MFC, Fig 1F) | Dehydrogenase Activity (MTT, Fig 1D) | |
| **2.23 μM ARI** | 0.9 | 0.9 | 1.0 | **0.8** | Fold change |
| | p = 0.9965 | p = 0.8570 | p = 0.4882 | **p = 0.0495** | p-value |
| **6 μM ARI** | 0.6 | 1.0 | 1.0 | **0.7** | Fold change |
| | p = 0.5420 | p = 0.9783 | p = 0.3374 | **p = 0.0002** | p-value |
| | Hepatotoxicity | | | | |
| | Membrane Leakage (LDH, Fig 2B)* | Apoptosis | | | |
| | | Execution (c-3/7 activity, Fig 2E) | Initiation (c-9 activity, Fig 2F) | | |
| **2.23 μM ARI** | 0.02 | 1.1 | 1.2 | | Fold change |
| | p = 0.6762 | p = 0.3789 | p = 0.5294 | | p-value |
| **6 μM ARI** | 0.04 | 1.1 | 1.0 | | Fold change |
| | p = 0.2459 | p = 0.2810 | p>0.9999 | | p-value |
| | Proliferation | Cell cycle | Senescence | | |
| | Ki67 protein (MFC, Fig 3C) | G0/G1 (MFC, Fig 4B) | ß-galactosidase (activity, Fig 5A) | | |
| **2.23 μM ARI** | 0.8 | **1.2** | 1.8 | | Fold change |
| | p = 0.0684 | **p = 0.0053** | p = 0.4070 | | p-value |
| **6 μM ARI** | **0.7** | **1.3** | **3.3** | | Fold change |
| | **p = 0.0067** | **p = 0.0002** | **p = 0.0054** | | p-value |

Fold change: differences of treated samples compared to the untreated control; statistically significant differences between the samples (Dunnett's test) are emboldened. *Membrane leakage range is between 0 (no leakage) and 1 (disrupted membrane). MFC: microcapillary flow cytometry; NR: neutral red assay; LDH: lactate dehydrogenase assay, c-3/7: caspase-3/7, c-9: caspase-9.

of cell cycle inhibitors, Cip/Kip [41]. We were unable to amplify *p16*, which encodes another cell cycle inhibitor of the p16/cyclin-dependent kinase (CDK) 4: cyclin D/retinoblastoma pathway, like the p21 and p27 [41]. Inability to amplify the *p16* gene in cell lines agrees with the fact that loss of p16 activity is the most common event in human tumorigenesis [42]. The ARI-treated cells are likely arrested before the G1/S transition. Due to the inability to amplify *p16* mRNA, we cannot prove whether the ARI-mediated cell cycle arrest is in G0 or G1 phase. Lack of *Gadd45α* induction indicates that there is no extensive DNA damage, however, it does not prove it. Gadd45α protein expression along with expression analyses of Gadd45β and Gadd45γ should be investigated in the future.

According to the report from Danish Poison Information Centre, ARI has one of the fewest metabolic side-effects among the second-generation antipsychotics [43]. Its single overdose has few and mild symptoms related to the sedative properties. We observed that ARI decreased liver cell division in concentrations that equaled plasma concentrations of patients when side-effects were reported, i.e. at the laboratory alert levels. The decreased division was first observed after two weeks of ARI treatment, when ARI levels also reach a steady-state in patients [9]. ARI elicited reduced hepatocyte regeneration could be important in the cases of alcohol and illegal drug abuse, when the ability of liver regeneration is crucial. 50% of the reported above-mentioned ARI-poisoned Danish patients were readmitted to hospitals because of the second incidence of drug poisoning within one year of the first ARI overdose

[43]. Also, both reported cases of the ARI-induced severe liver toxicity (DILI) occurred in patients with a history of alcohol and cocaine abuse [4, 5]. Severe liver toxicity was also seen in rats when they were co-treated with a combination of ARI and fluvoxamine or carbamazepine [25]. Fluvoxamine was reported as hepatotoxic [44], and according to the LiverTox database, hepatotoxicity of carbamazepine is uncommon but well described [45].

Reduced liver cell division because of continuous ARI treatment, which we are describing here for the first time, is the possible mechanism underlying a decreased ability of liver regeneration that may result in DILI, in a compromised liver. As alcohol and illicit drugs are comorbidities in up to half of patients with schizophrenia and ARI is being clinically tested for addiction reduction, clinicians need to be aware of lowered liver cell regeneration. Monitoring liver functions is, therefore, important when ARI is prescribed in patients with a history of alcohol and drug-abuse.

## Supporting information

**S1 Fig.** Population and viability profiles of Fao cells corresponding to Fig 1B and 1F, left panels.
(TIF)

**S2 Fig.** Population and viability profiles of hepatocytes corresponding to Fig 1B and 1F, right panels.
(TIF)

**S3 Fig. a)** Population and proliferation profiles corresponding to Fig 3B. **b)** Population and proliferation profiles corresponding to Fig 3C.
(TIF)

**S4 Fig. a)** Population and DNA content profiles corresponding to Fig 4A and 4C. **b)** Population and DNA content profiles corresponding to Fig 4B and 4D.
(TIF)

**S5 Fig. Summary statistics of data on Fao cells: Brown-Forsythe test and ANOVA.**
(TIF)

## Acknowledgments

We are grateful to Dr Ángela M. Valverde, Instituto de Investigaciones Biomedicas 'Alberto Sols', CSIC, Madrid, Spain for a kind donation of mouse immortalized hepatocytes through H2020-MSCA-ITN:721236 TREATMENT project.

## Author Contributions

**Conceptualization:** Irina Milisav.

**Investigation:** Tinkara Pirc Marolt, Barbara Kramar, Klara Bulc Rozman.

**Resources:** Dušan Šuput.

**Supervision:** Irina Milisav.

**Visualization:** Tinkara Pirc Marolt, Barbara Kramar, Klara Bulc Rozman.

**Writing – original draft:** Irina Milisav.

**Writing – review & editing:** Tinkara Pirc Marolt, Dušan Šuput, Irina Milisav.

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
