## [Decision Letter · Decision Letter 0]

2 Aug 2020

PONE-D-20-18998

Aripiprazole reduces liver cell division

PLOS ONE

Dear Dr. Milisav,

Thank you for submitting your manuscript to PLOS ONE. After careful consideration, we feel that it has merit but does not fully meet PLOS ONE’s publication criteria as it currently stands. Therefore, we invite you to submit a revised version of the manuscript that addresses the points raised during the review process.

Both reviewers and AE feel there is merit in the content of the Ms. However major issues need to be addressed before it can be re-considered for publication

We look forward to receiving your revised manuscript.

Kind regards,

Ezio Laconi, MD, PhD

Academic Editor

PLOS ONE

Journal Requirements:

2. In table 1, please provide the actual numerical results, including fold change and p-values.

3. We note that in line 71 you refer to schizophrenic vs. healthy populations. Please be more specific about what "healthy" population means. For example, does this mean non-schizophrenic and/or no other major underlying health conditions?

Reviewers' comments:

Reviewer's Responses to Questions

**Comments to the Author**

1. Is the manuscript technically sound, and do the data support the conclusions?

Reviewer #1: Partly

Reviewer #2: Yes

2. Has the statistical analysis been performed appropriately and rigorously? 

Reviewer #1: No

Reviewer #2: Yes

3. Have the authors made all data underlying the findings in their manuscript fully available?

Reviewer #1: Yes

Reviewer #2: Yes

4. Is the manuscript presented in an intelligible fashion and written in standard English?

Reviewer #1: No

Reviewer #2: Yes

5. Review Comments to the Author

Reviewer #1: Comments to the Authors

The manuscript from Tinkara Pirc Marolt et al. entitled “Aripiprazole reduces liver cell division” (Manuscript Number: PONE-D-20-18998) seems to be an interesting manuscript. There are points which need to be addressed.

1. Abstract appears so long. It can be short and concise.

2. Lines 194-, figures do not appear in order: Figs 1a, b, 1d, 1e, 1c, and 1f.

3. Lines 212-, the word “Untr” is not seen in the figure.

4. Table 1 seems difficult to be understood by readers. It has quite a bit of room for improvement.

5. In Table 1, cell counts are less than the control in both 2.23 and 6 uM ARI. There may be difference comparing the result of Fig. 1b.

6. In terms of statistics, authors used ANOVA to compare groups, but a test for normality seems lacking.

Reviewer #2: In this paper, it is interesting that aripiprazole reduces the number of hepatocytes by inhibiting the division of hepatocytes. However, there are still some problems in the experimental design and discussion.

Majoy modification suggestions:

1. It is suggested that primary hepatocytes of aripiprazole induced liver injury animal model be used to verify the conclusion that the number of hepatocytes decreased but the activity remained unchanged.

2. To increase the research on signal pathway of aripiprazole inhibiting hepatocyte division.

Minor modification comments:

1. Flow cytometry results on cell cycle should be displayed, not just statistical charts.

2. The morphological changes of aging hepatocytes should be observed.

3. Line197-198：The conclusion of Figure1c should be added. For example: ‘this indicates that the cytotoxicity is the same after long-term treatment’ should be added later. Meanwhile, since Figure1d-e are described first in this paper, Figure1c should be placed after Figure1d-e (i.e., the Ordinal Numbers of Figure1d-e should be changed to Figure 1c and 1d, and the Ordinal Numbers of Figure1c should be changed to Figure1e).

4. Line 198-199：The method used here is the ‘capillary flow cytometry’ but the method described in line 210 is the ‘microcapillary cytometry.

5. Line 190-199：The results of this section should be summarized in a sentence at the end of the paragraph.

6. Line251-253：The description of Figure4 is too simple and should be more detailed.

7. In the discussion part, the experimental contents should be summarized and analyzed in depth.

8. Subheadings should be added to each section of the results to summarize what the results reveal so that the reader can better frame the text.

6. PLOS authors have the option to publish the peer review history of their article (what does this mean?). If published, this will include your full peer review and any attached files.

Reviewer #1: No

Reviewer #2: No

---

## [Author Response · Author response to Decision Letter 0]

16 Sep 2020

Dear Dr Laconi, dear Reviewers,

thank you for your valuable comments towards the manuscript improvement. We have addressed all raised points and hope that our manuscript is now suitable for publication.

Journal requirements

Done.

2. In table 1, please provide the actual numerical results, including fold change and p-values.

Provided. 

3. We note that in line 71 you refer to schizophrenic vs. healthy populations. Please be more specific about what "healthy" population means. For example, does this mean non-schizophrenic and/or no other major underlying health conditions?

Non-schizophrenic population (line 64 of the manuscript was modified accordingly).

The text now reads:” Alcohol or illicit drug dependence is estimated in up to 50 % of patients with schizophrenia [6, 7], and is 4.6-times higher than in the population without this disease [8]” 

Review Comments to the Author

Reviewer #1: Comments to the Authors

The manuscript from Tinkara Pirc Marolt et al. entitled “Aripiprazole reduces liver cell division” (Manuscript Number: PONE-D-20-18998) seems to be an interesting manuscript. There are points which need to be addressed.

1. Abstract appears so long. It can be short and concise.

The abstract was shortened.

2. Lines 194-, figures do not appear in order: Figs 1a, b, 1d, 1e, 1c, and 1f.

The figures were re-numbered.

3. Lines 212-, the word “Untr” is not seen in the figure.

Thank you, removed.

4. Table 1 seems difficult to be understood by readers. It has quite a bit of room for improvement. 

The actual numerical results, including fold change and p-values were added.

5. In Table 1, cell counts are less than the control in both 2.23 and 6 uM ARI. There may be difference comparing the result of Fig. 1b.

There is no discrepancy between the data in Table1 and Fig 1b (now left panel). The number of long-term treated cells was set equal at the day 0 (100 %). The numbers of cells in each sample were counted 2 and 4 days later and the numbers of ARI treated cells in every biological replicate were lower than that of the cells in any other sample in the case of the 6 uM ARI treatment in Fao and immortalized hepatocytes.

6. In terms of statistics, authors used ANOVA to compare groups, but a test for normality seems lacking.

Data were analysed with GraphPad Prism, which has in built algorithms that test the equality of variances from medians, the Brown–Forsythe test. The data were added as a supplement S5.

Reviewer #2: In this paper, it is interesting that aripiprazole reduces the number of hepatocytes by inhibiting the division of hepatocytes. However, there are still some problems in the experimental design and discussion.

Major modification suggestions:

1. It is suggested that primary hepatocytes of aripiprazole induced liver injury animal model be used to verify the conclusion that the number of hepatocytes decreased but the activity remained unchanged.

We agree that the primary hepatocytes are the best model, however, they do not proliferate in cell cultures. An animal experiment would require an application to the ethics committee to perform an invasive procedure, liver regeneration after hepatectomy. ARI and OLA are hydrophobic and one would need to feed the animals by gavage.

We have additionally confirmed the reduced growth upon the ARI treatment on the neonatal immortalized hepatocytes (Figure 1b, right panel).

2. To increase the research on signal pathway of aripiprazole inhibiting hepatocyte division.

Gene expression of DNA-damage induced genes (p53 and Gadd45a) and cell cycle inhibitors (p16, p21 and p27) was determined.

Minor modification comments:

1. Flow cytometry results on cell cycle should be displayed, not just statistical charts.

These are in supplemental figures S1-S4.

2. The morphological changes of aging hepatocytes should be observed.

Morphology of long-term treated cells is depicted in Fig 1a. We have not observed nor measured any increase in cell demise.

3. Line197-198：The conclusion of Figure1c should be added. For example: ‘this indicates that the cytotoxicity is the same after long-term treatment’ should be added later. Meanwhile, since Figure1d-e are described first in this paper, Figure1c should be placed after Figure1d-e (i.e., the Ordinal Numbers of Figure1d-e should be changed to Figure 1c and 1d, and the Ordinal Numbers of Figure1c should be changed to Figure1e).

The Figures 1c-1e were re-numbered and the conclusion was added: “This indicates that cell survival is the same in all treated cells during the long-term treatment.”

4. Line 198-199：The method used here is the ‘capillary flow cytometry’ but the method described in line 210 is the ‘microcapillary cytometry.

The term ‘microcapillary flow cytometry’ is now used.

5. Line 190-199：The results of this section should be summarized in a sentence at the end of the paragraph.

“This indicates that cell survival is the same in all treated cells during the long-term treatment.”

6. Line251-253：The description of Figure4 is too simple and should be more detailed.

The figure and its legend are supplemented.

7. In the discussion part, the experimental contents should be summarized and analyzed in depth.

New text was added.

8. Subheadings should be added to each section of the results to summarize what the results reveal so that the reader can better frame the text.

Subheadings are added.

---

## [Decision Letter · Decision Letter 1]

2 Oct 2020

Aripiprazole reduces liver cell division

PONE-D-20-18998R1

Dear Dr. Milisav,

We’re pleased to inform you that your manuscript has been judged scientifically suitable for publication and will be formally accepted for publication once it meets all outstanding technical requirements.

Kind regards,

Ezio Laconi, MD, PhD

Academic Editor

PLOS ONE

Additional Editor Comments (optional):

Reviewers' comments:

Reviewer's Responses to Questions

**Comments to the Author**

1. If the authors have adequately addressed your comments raised in a previous round of review and you feel that this manuscript is now acceptable for publication, you may indicate that here to bypass the “Comments to the Author” section, enter your conflict of interest statement in the “Confidential to Editor” section, and submit your "Accept" recommendation.

Reviewer #1: All comments have been addressed

Reviewer #2: All comments have been addressed

2. Is the manuscript technically sound, and do the data support the conclusions?

Reviewer #1: Yes

Reviewer #2: Yes

3. Has the statistical analysis been performed appropriately and rigorously? 

Reviewer #1: Yes

Reviewer #2: Yes

4. Have the authors made all data underlying the findings in their manuscript fully available?

Reviewer #1: Yes

Reviewer #2: Yes

5. Is the manuscript presented in an intelligible fashion and written in standard English?

Reviewer #1: Yes

Reviewer #2: Yes

6. Review Comments to the Author

Reviewer #1: Comments to the Authors

According to the comments from the Reviewers, the Authors responded adequately and conducted several modifications appropriately. This seems a quite well-written and reshaped manuscript. Therefore, this can be suitable for publication in the journal.

Reviewer #2: After the revision, the structure of the paper is reasonable and logical. The results are illustrative and the statistical analysis is performed properly and strictly. The manuscript is easy to understand and written in standard English. In general, the conclusions are presented in an appropriate manner and supported by data. Therefore, my recommendation is acceptance.

7. PLOS authors have the option to publish the peer review history of their article (what does this mean?). If published, this will include your full peer review and any attached files.

Reviewer #1: No

Reviewer #2: No

---

## [Editor Report · Acceptance letter]

16 Oct 2020

PONE-D-20-18998R1 

Aripiprazole reduces liver cell division 

Dear Dr. Milisav:

I'm pleased to inform you that your manuscript has been deemed suitable for publication in PLOS ONE. Congratulations! Your manuscript is now with our production department. 

Kind regards, 

on behalf of

Dr. Ezio Laconi 

Academic Editor

PLOS ONE